# An Entropy-Based Algorithm with Nonlocal Residual Learning for Image Compressive Sensing Recovery

**DOI:** 10.3390/e21090900

**Published:** 2019-09-17

**Authors:** Zhonghua Xie, Lingjun Liu, Cui Yang

**Affiliations:** 1School of Information Science and Technology, Huizhou University, Huizhou 516007, China; eezhxie@gmail.com; 2School of Electronic and Information Engineering, South China University of Technology, Guangzhou 510641, China; yangcui@scut.edu.cn

**Keywords:** compressive sensing (CS), residual leaning, group sparsity, Laplacian scale mixture (LSM), relative entropy, denoising-based approximate message passing (D-AMP)

## Abstract

Image recovery from compressive sensing (CS) measurement data, especially noisy data has always been challenging due to its implicit ill-posed nature, thus, to seek a domain where a signal can exhibit a high degree of sparsity and to design an effective algorithm have drawn increasingly more attention. Among various sparsity-based models, structured or group sparsity often leads to more powerful signal reconstruction techniques. In this paper, we propose a novel entropy-based algorithm for CS recovery to enhance image sparsity through learning the group sparsity of residual. To reduce the residual of similar packed patches, the group sparsity of residual is described by a Laplacian scale mixture (LSM) model, therefore, each singular value of the residual of similar packed patches is modeled as a Laplacian distribution with a variable scale parameter, to exploit the benefits of high-order dependency among sparse coefficients. Due to the latent variables, the maximum a posteriori (MAP) estimation of the sparse coefficients cannot be obtained, thus, we design a loss function for expectation–maximization (EM) method based on relative entropy. In the frame of EM iteration, the sparse coefficients can be estimated with the denoising-based approximate message passing (D-AMP) algorithm. Experimental results have shown that the proposed algorithm can significantly outperform existing CS techniques for image recovery.

## 1. Introduction

Compressive sensing (CS) [1,2] has drawn quite an amount of attention as a novel digital signal sampling theory when the signal is sparse in some domain. It performs signal acquisition and processing using far fewer samples than required by the Nyquist rate. Breakthroughs in CS have the potential to greatly reduce the sampling rates in numerous signal processing applications, such as cameras, medical scanners, radar imaging, and fast analog to digital converters. The measurement vector of a signal is obtained by a linear system through multiplying the signal by a known measurement matrix, which is usually constructed from a Gaussian or Bernoulli random operator. If the signal that contains few nonzero components is sparse under some proper basis, CS looks for the sparsest solution of the underdetermined system to get an accurate reconstruction. Finding a sparse representation can be achieved using greedy algorithms such as matching pursuit (MP) and orthogonal matching pursuit (OMP) [3,4] or iterative shrinkage–thresholding algorithms and their variations such as fast iterative shrinkage–thresholding algorithm (FISTA), a shorthand for Nesterov’s algorithm (NESTA) [5,6], and Bayesian compressive sensing or sparse Bayesian learning (SBL) algorithms [7,8].

Owing to the fact that exploiting a prior knowledge of the original signals plays a critical role in the performance of compressive sensing reconstruction, much efforts have been made to develop effective regularization terms or signal sparse models. However, most of the conventional CS reconstruction methods exploit a set of fixed bases (e.g., Discrete Cosine Transform (DCT) [9], wavelets [10], total variation (TV) [11,12], and learned dictionary [13,14]) for the entirety of signals. Unfortunately, these methods, which are irrespective of the nonstationarity of natural signals and cannot achieve a high enough degree of sparsity are less appropriate for many imaging applications. For instance, CS reconstruction methods based on TV exploit the sparsity of image in gradient domain, and the resulting convex optimization problems can be efficiently solved by the class of surrogate-function-based methods, but these methods based on global sparsity favor piecewise constant structures and hence make the image details become over-smoothed. The sparse model based on learned dictionary [14] assumes that each patch of image can be accurately represented by a few elements from a basis set called dictionary, which is learned from natural images, nonetheless, the patch-based sparse representation model of natural images usually suffers from some limits, such as dictionary learning with great computational complexity, ignoring the relationship among sparse coefficients.

To rectify these problems, many works have incorporated additional prior knowledge about transform coefficients into the CS recovery framework, such as Gaussian scale mixtures (GSM) models [15], group sparsity [16], tree-structured wavelet [17,18], and nonlocal sparsity [19,20]. By modeling statistical dependencies among sparse coefficients, one can greatly reduce the uncertainty about the unknown signal, resulting in more accurate reconstruction. For instance, the structure of the wavelet coefficients is modeled by the hidden Markov (HM) model in [18] to exploit the tree sparsity of image, and the turbo scheme that alternates between inference on the HM tree structure with standard belief propagation and inference on the compressive sensing measurement structure with the generalized approximate message passing algorithm is applied, leading to improvements in performance, but it is based on global sparsity and thus limits its improvement. More recently, deep learning-based methods that use no hand-designed and end-to-end training models learn how to best use the structure within the external data with deep learning networks [21]. Either pure convolutional layers (Deep Inverse [22]) or a combination of convolutional and fully connected layers (DR2-Net [23] and ReconNet [24]) is used to build deep learning frameworks. Deep neural network architectures that combine the model-based method and the data-driven method have been developed in [25,26]. The idea of Generative Adversarial Networks (GAN) is used in magnetic resonance imaging (MRI) recovery in [27,28] to reduce the aliasing artifacts. The generative network uses the u-net, while the discriminative network uses the Deep Convolution GAN network. The deep CS method has been proposed to recover images with uncertainty measurement of MRI in [29]. Unfortunately, these methods are held back by the fact that there exists almost no theory governing their performance and they take a lot of time and vast amounts of data to train.

Nonlocal sparsity, which refers to the fact that a patch often has many nonlocal similar patches to it across the image, has shown most beneficial to CS image recovery. The nonlocal total variation (NLTV) regularization model for CS image recovery has been proposed in [19] by using the self-similarity property in gradient domain. In order to obtain the adaptive sparsity regularization term for CS image recovery process, the local piecewise autoregressive model is designed in [20]. In [30], similar patches are grouped to form a two-dimensional data matrix for characterizing the low-rank property, leading to the CS recovery method via nonlocal low-rank regularization (NLR-CS). In [31,32], the probabilistic graphical model is established, which uses collaborative filtering [33] to promote sparsity of packed patches. Meanwhile, composite sparse models are developed to soften and complement the nonlocal sparsity for irregular structures, so as to preserve image details. A new framework for image compressive sensing recovery via structural group sparse representation (SGSR) modeling is proposed in [34], which enforces image sparsity and self-similarity simultaneously under a unified framework in an adaptive group domain. In [35], two sets of complementary ideas for regularizing image reconstruction process are proposed, the sparsity regularization parameters are locally estimated for each coefficient and updated along with adaptive learning of PCA-based dictionaries. The nonlocal self-similarity constraint is introduced into the overall cost functional to improve the robustness of the model. In [36], the CS reconstruction algorithm takes both the low-rank constraint of similar image patches and the bilateral filter constraint as the joint prior information of natural images to enhance the recovery effect of image textures and edges, thus improving the performance of the CS algorithm. In [37], the low-rank and total variation regularizations are jointly utilized for MR image super-resolution. Despite the steady progress in CS methods based on nonlocal sparsity, they still tend to smooth the detailed image textures, degrading the image visual quality, for the reason that the lack of self-repetitive structures and noise corruption for data is unavoidable.

In this work, we propose a novel prior model for image CS reconstruction by residual sparse learning. To improve the performance of nonlocal sparse-based image reconstruction, the concept of residual sparsity of similar packed patches is proposed, and thus, the problem of image CS reconstruction is turned into one that reduces the residual. In order to reduce the residual, we first obtain some good estimation of the original image by the Block-Matching and 3D filtering (BM3D) method [33], and then centralize the sparse coefficients of the intermediate noisy image to the estimation. Through the Bayesian method, we can determine the adaptive parameters of this optimization problem. More specifically, the residual is represented by Laplacian scale mixture (LSM) [38] models, which are usually adopted to model the sparse coefficients. Each singular value of similar residual matrices packed and rearranged by similar patches of the intermediate noisy image and the pre-estimations is modeled as a Laplacian distribution with a variable scale parameter, resulting in weighted singular value minimization problems, where weights are adaptively assigned according to the signal-to-noise ratio. To solve this model, the expectation–maximization (EM) [39] method with a loss function of relative entropy is adopted, turning the CS recovery problem into a prior information estimation problem and a singular value minimization problem. Specially, owing to its promising performance and efficiency, we are motivated to apply the denoising-based approximate message passing (D-AMP) algorithm, which is an iterative algorithm that can be used in signal and image reconstruction by performing denoising at each iteration [40], to solve the latter. Experimental results on natural images show that our approach can achieve more accurate reconstruction than other competing approaches.

## 2. Background

### 2.1. Compressed Sensing

Compressive sensing (CS) is a signal acquisition technique that enables accurate reconstructions of images from far fewer measurements, acquired by linear projection (i.e., *y* = *Ax* + *w*) than the number of unknowns, where x∈ℂn is the original signal, y∈ℂm is the measurement vector, A∈ℂm×n is the measurement matrix, and *w* denotes the additive noise. It is an underdetermined linear system because of m<n. When the underlying signal is sufficiently sparse in some transform domain or dictionary, this underdetermined problem can be solved by greedily tracing the sparsest solution with greedy-based algorithms, iteratively denoising the intermediate noisy signal with iterative thresholding algorithms, or Bayesian inferences with sparse Bayesian learning algorithms. The image reconstruction problem in CS is often formulated as follows:(1)x^=argminx12‖y−Ax‖22+λℜ(x)

The first term is the data fidelity term that represents the closeness of the solution to the measurements. The second term is a regularization term that represents a priori sparse information of the original signal. λ is a regularization parameter that balances the contribution of both terms. In (1), x is a vectorized version of the image to be reconstructed, y denotes the imaging measurements, A is the sensing or measurement matrix for the application. In a single pixel camera, A is a sequence of 1s and 0s representing the modulation of a micromirror array, while in a compressively sampled MRI, A might be samples of an n×n Fourier matrix. CS recovery methods exploit the sparsity of signal in some transform domain or dictionary, such as wavelets, total variation (TV), and learned dictionary, leading to various forms of ℜ(x) (i.e., ${\left\|\Psi x\right\|}_{1}$, ${\left\|Dx\right\|}_{1}$ (using a convex relaxation of the *ℓ*_0_ counting “norm” for sparsity), and ${\left\|x\right\|}_{TV}$ respectively).

### 2.2. CS Recovery Based on Nonlocal Sparsity

As shown in Figure 1, self-repeating patterns are abundant in natural image, which can be characterized by the nonlocal sparsity. Most nonlocal regularization models consist of two components: patch grouping, for characterizing self-similarity of images and group sparse approximation, for sparsity enforcement. As shown in Figure 1, patches in images are grouped and rearranged to form low-rank matrices. More specifically, for each local patch we can find the first *M* most similar nonlocal patches to it. In practice, this can be done by block matching based on Euclidean distance within a local window of size F×F. Let Rix (or xi) denote an exemplar patch located at the *i*-th position, which is the vectorized version of the patch. Patches that are similar to Rix are found to form the matrix Xi=[Ri1x,Ri2x,…,RiNx], Xi∈ℝM×N, *M* ≥ *N*, including Rix itself. Because the patches are similar, the formed matrix has a low-rank property. An objective function that reflects the group sparsity of similar patches with a low-rank regularization term for CS recovery can be formulated as follows:(2)x^=argminx12‖y−Ax‖22+μ∑i=1G{‖Xi−Li‖F2+λ‖Li‖∗}
where μ and λ are two regularization parameters, *G* is the total number of similar patch groups, Li is the low-rank data matrices to be estimated; ‖Li‖∗ is the nuclear norm of Li, taking a sum value of its singular values, therefore, ‖Li‖∗=∑j|li,j|, li=[li,1,li,2,…,li,N] denotes the singular value vector of Li. 

### 2.3. Denoising-Based Approximate Message Passing

The minimizing problems (e.g., Equations (1) and (2)) can be solved by iterative optimization algorithms, such as iterative shrinkage–thresholding methods [5,41], alternating direction method of multipliers (ADMM) [42], or Bregman iterative algorithms [43]. The approximate message-passing reconstruction algorithm defined by Donoho et al. [40], based on the theory of belief propagation in graphical models, has recently become a popular algorithm for solving signal reconstruction problems in linear systems, as defined in Equation (1). By employing quadratic approximation, the belief propagation is turned into a simple version with less computation [44]. The final alternating expressions in the approximate message passing (AMP) algorithm to solve $\underset{x}{\min}{\left\|y-Ax\right\|}_{2}^{2}/2+\lambda {\left\|\Psi x\right\|}_{1}$ where Ψx denotes the wavelet transform of x are
(3)$$x^{\left(t+1\right)}=\eta (x^{\left(t\right)}+A^{*}z^{\left(t\right)})$$
(4)$$z^{\left(t\right)}=y-Ax^{\left(t\right)}+\frac{z^{\left(t-1\right)}}{m}{\left\|\eta^{\prime }(x^{\left(t-1\right)}+A^{*}z^{\left(t-1\right)})\right\|}_{1}$$
where $x^{\left(t\right)}$ and $z^{\left(t\right)}$ are the estimates of *x* and the residual at iteration *t*. The iteration starts from $x^{\left(0\right)}$= 0, $z^{\left(0\right)}$= *y*. $A^{*}$ is the conjugate transpose of *A*. The functions η(⋅) and $\eta^{\prime }(\cdot )$ are the wavelet threshold function and its first derivative, respectively. The last term in Equation (4) is called the “Onsager reaction term” [19] in statistical physics. This Onsager reaction term helps improve the phase transition (trade-off between the measurement rate and signal sparsity) of the reconstruction process over existing iterative shrinkage–thresholding algorithms [5,41]. Some AMP variants [45,46] have been proposed with various forms of ℜ(x), such as total variation and the Cauchy prior in wavelet domain [46]. The AMP algorithm can be summarized in three steps: the residue update step (i.e., Equation (4)), the back-projection step to yield a noisy image, and the proximal denoising correction (i.e., Equation (3)). As a result, the AMP algorithm can be turned into the denoising-based approximate message passing algorithm [31]. In [31], each denoiser is treated as a black box to estimate the original signal through the denoising of the intermediate noisy image. Thus, instead of assuming any knowledge of the signal information, the D-AMP algorithm employs the denoising algorithm to achieve its goal, which makes the D-AMP algorithm applicable to a wide variety of signal classes and a wide variety of denoisers. The D-AMP algorithm employs a denoiser in the following iteration: (5)$$x^{\left(t+1\right)}=D(x^{\left(t\right)}+A^{*}z^{\left(t\right)})$$
(6)$$z^{\left(t\right)}=y-Ax^{\left(t\right)}+\frac{z^{\left(t-1\right)}}{m}D^{\prime }(x^{\left(t-1\right)}+A^{*}z^{\left(t-1\right)})$$
where D(⋅) denotes the denoising operator. Note that it is not easy to compute the Onsager term, because it involves computing the derivative $D^{\prime }(\cdot )$, and D(⋅) do not have an explicit input–output relation. In [30], the Monte Carlo (MC) method [47] is utilized to simulate $D^{\prime }(\cdot )$ with random numbers. Note that the intermediate noisy image is defined by $h^{\left(t\right)}=x^{\left(t\right)}+A^{*}z^{\left(t\right)}$ as the input of the denoising operator D(⋅).

## 3. Image CS Recovery via Nonlocal Residual Learning and D-AMP

### 3.1. Residual Learning

Image nonlocal self-similarity has been widely adopted in patch-based image CS reconstruction methods. Despite the great success, most of the existing works exploit the nonlocal sparsity by enforcing the packed patches to be as similar as possible, resulting in over-smoothed outcomes. In fact, unlike large-scale edges, the fine-scale textures have much higher randomness in local structure, and they are hard to characterize by using existing nonlocal models. In this paper, we propose a novel nonlocal model with residual learning. We first obtain some good estimation of the original image by the BM3D method, and then centralize the sparse coefficients of the intermediate noisy image to the estimation. Following the D-AMP algorithm, we first propose a denoising model as follows:(7)$${\hat{A}}_{i}=\underset{A_{i}}{\arg\min}{\left\|H_{i}-D_{i}A_{i}\right\|}_{F}^{2}+\lambda {\left\|D_{i}(A_{i}-B_{i})\right\|}_{*}$$
where ${\left\|\cdot \right\|}_{F}$ is the Frobenius norm, $H_{i}$ is the low-rank data matrix of the noisy image, $A_{i}$ is the group sparse coefficients of $H_{i}$, $B_{i}$ denotes the true group sparse coefficients, $D_{i}$ is an orthogonal dictionary, which is obtained by performing singular value decomposition (SVD) on $H_{i}$, and λ is a regularization parameter. $B_{i}$ cannot be obtained in practice, nonetheless, we can compute some good estimation of it by the BM3D method. The residual *S* is defined as
(8)S=D(A−B)

Instead of pursuing a smoothed B with ${\left\|D_{i}A_{i}\right\|}_{*}$, A is encouraged to be with a little noise in Equation (7), thus, the results in favor of preserving details. 

The optimization problem in Equation (7) is a nuclear norm proximal problem [48], which can be easily solved in closed-form by imposing a soft thresholding operation on the singular values of the observation matrix (i.e., by taking a derivative with respect to $A_{i}$):(9)$${\hat{A}}_{i}=S_{\frac{\lambda }{2}}(D_{i}^{T}H_{i}-B_{i})+B_{i}$$
where $H_{i}=U\sum V^{T}$ is the SVD of $H_{i}$. When $H_{i}$ is a square matrix, U and V come out to be the principal component analysis (PCA) dictionary D. $S_{\frac{\lambda }{2}}(D_{i}^{T}H_{i}-B_{i})$ is the soft thresholding function on the vectorized version of the sparse coefficients $D_{i}^{T}H_{i}-B_{i}$ with parameter $\frac{\lambda }{2}$. We substitute our denoising method for the denoising operator D(⋅) in Equations (5) and (6), and finally obtain our CS recovery method by combining with the D-AMP algorithm. However, the regularization parameter λ is hard to determine, which might be the key to the success of CS recovery, thus we employ the Laplacian scale mixture model to represent the sparse coefficients of the residual, and then use the Bayesian inference to adaptively determine the parameters.

### 3.2. LSM Prior Modeling

The regularization parameter in variational image restoration is intrinsically connected with the shape parameter of sparse coefficients’ distribution in Bayesian methods. How to set those parameters in a principled yet spatially adaptive fashion turns out to be a challenging problem, especially for the class of nonlocal image models. To model the sparse coefficients of the residual with the Laplacian scale mixture distribution, we first offer some definitions about LSM.

A random variable $\theta_{i}=\kappa_{i}^{-1}\varsigma_{i}$ is a Laplacian scale mixture if $\varsigma_{i}$ has a Laplacian distribution with scale 1 (i.e., $p(\varsigma_{i})=\exp(-|\varsigma_{i}|)/2$), and the multiplier variable $\kappa_{i}$ is a positive random variable with probability $p(\kappa_{i})$ [38]. Supposing that $\varsigma_{i}$ and $\kappa_{i}$ are independent, conditioned on the parameter $\kappa_{i}$, the coefficient $\theta_{i}$ has a Laplacian distribution with inverse scale $p(\theta_{i}|\kappa_{i})=\kappa_{i}\exp(-\kappa_{i}|\theta_{i}|)/2$. The distribution over $\theta_{i}$ is therefore a continuous mixture of Laplacian distributions with different inverse scales:(10)$$p(\theta_{i})=\int_{0}^{\infty }p(\theta_{i}|\kappa_{i})p(\kappa_{i})d\kappa_{i}=\int_{0}^{\infty }\frac{\kappa_{i}}{2}\exp(-\kappa_{i}|\theta_{i}|)p(\kappa_{i})d\kappa_{i}$$
The distribution in Equation (10) is defined as a Laplacian scale mixture. Note that for most choices of $p(\kappa_{i})$, we do not have an analytical expression for $p(\theta_{i})$.

We formulate the LSM prior model, and apply the maximum a posteriori (MAP) theory to estimate the original signal. Let *a*, *b*, and *h* represent the vectorized version of *A*, *B*, and *H* respectively. Using Bayesian formula, we might derive the following MAP estimation problem
(11)$$\hat{a}=\arg\underset{a}{\min}p(a|h)=\arg\underset{a}{\min}p(h|a)p(a)$$

According to Equation (11), we need to define the likelihood term p(h|a) and the prior distribution p(a). First, the additive noise *v* is assumed to be white Gaussian with variance $\sigma^{2}$, therefore, $v\sim N(0,\sigma^{2})$. Thus, we have the following likelihood
(12)$$p(h|a)=\frac{1}{{\left(2\pi \sigma^{2}\right)}^{n/2}}\exp(-\frac{1}{2\sigma^{2}}{\left\|h-Da\right\|}_{2}^{2})$$

Second, we characterize the nonlocal sparsity of image with the Laplacian scale mixture model. Let $s_{i}=[s_{i,1},s_{i,2},\ldots ,s_{i,N}]$ be the singular value vector of the low-rank matrix $S_{i}=D_{i}^{T}(A_{i}-B_{i})$. For each coefficient $s_{i,j}$, we assign it a Laplacian distribution with a variable scale parameter,
(13)$$p(s_{i,j})=\int_{0}^{\infty }p(s_{i,j}|\gamma_{i,j})p(\gamma_{i,j})d\gamma_{i,j}=\int_{0}^{\infty }\frac{\gamma_{i,j}}{2}\exp(-\gamma_{i,j}|s_{i,j}|)p(\gamma_{i,j})d\gamma_{i,j}$$
with a Gamma distribution prior over the scale parameter (i.e., $\gamma_{i,j}\sim \mathrm{Gamma}(\alpha ,\beta )$). The observation that the distribution of singular values of the residual can be modeled by a Laplacian was proposed and validated in [49]. Here, we extend this idea by viewing the scale parameter as a random variable, for achieving a better spatial adaptation. Assuming that $s_{i,j}$ are i.i.d, then the LSM prior of the sparse coefficient *s* can be expressed as $p(s)=\prod_{i=1}^{G}\prod_{j=1}^{N}p(s_{i,j})$.

The proposed hierarchical model is summarized in Figure 2. We will have an objective function that can be maximized with respect to *x*, if we observe the latent variable γ. The standard approach in machine learning when confronted with such a problem is the EM algorithm. Note that once we determine the spare coefficient vector *s*, the spare coefficient matrix $A_{i}$ can be obtained using the equation $S_{i}=D_{i}^{T}(A_{i}-B_{i})$.

### 3.3. Entropy-Based Algorithm for CS Recovery

We design an algorithm for CS Recovery with entropy theory [50]. We simultaneously learn the hidden parameters and do inference. To accomplish this task, the D-AMP algorithm is embedded within an EM framework. We design a loss function with the relative entropy of *Q* to a denoted by J(Q,a) (i.e., the Kullback–Leibler divergence). With Jensen’s inequality and according to Equation (11), we can obtain the upper bound on the posterior likelihood
(14)$$-\log p(s|h)\leq -\log p(h|s)-\int_{\gamma }Q(\gamma )\log\frac{p(s,\gamma )}{Q(\gamma )}d\gamma \;\;:=J(Q(\gamma ),a(s))$$
where,
(15)$$\log p(s,\gamma )=\log p(s|\gamma )+\log p(\gamma )=\sum_{i=1}^{G}\sum_{j=1}^{N}\left(-\gamma_{i,j}|s_{i,j}|+\log\frac{\gamma_{i,j}}{2}\right)+\log p(\gamma )$$

Performing coordinate descent in the function J(Q,a) leads to the following updates that are usually called the E step and the M step.
(16)$$Q^{\left(t+1\right)}=\arg\underset{Q}{\min}J(Q,a^{\left(t\right)})$$
(17)$$a^{\left(t+1\right)}=\arg\underset{a}{\min}J(Q^{\left(t+1\right)},a)$$

The E step and the M step represent the prior learning and the signal sparse coefficient reconstruction, respectively. First, we have equality in the Jensen inequality if Q(γ)=p(γ|s), which implies that the E step reduces to $Q^{\left(t+1\right)}=p(\gamma |s^{\left(t\right)})$. Second, let τ denote the expectation with respect to Q(γ), therefore, $\tau_{i,j}=\int_{\gamma_{i,j}}\gamma_{i,j}Q(\gamma_{i,j})d\gamma_{i,j}$. The M Step (17) simplifies to
(18)$$a_{i}^{\left(t+1\right)}=\arg\underset{a_{i}}{\min}\frac{{\left\|h_{i}^{\left(t+1\right)}-Da_{i}\right\|}_{2}^{2}}{2\sigma^{2}}+\sum_{j=1}^{N}\tau_{i,j}^{\left(t+1\right)}|s_{i,j}|$$
Finally, we have the proposed objective function for residual learning. These two teams are the data fidelity term and the nonlocal regularization, respectively.

The Gamma distribution and Laplacian distribution are conjugate, therefore, the posterior probability of $\gamma_{i,j}$ given $s_{i,j}$ is also a Gamma distribution with parameters α+1 and $\beta +|s_{i,j}|$. Hence, the expectations are given by
(19)$$\tau_{i,j}^{\left(t+1\right)}=\int_{\gamma_{i,j}}\gamma_{i,j}p(\gamma_{i,j}|s_{i,j}^{\left(t\right)})d\gamma_{i,j}=\frac{\alpha +1}{\beta +|s_{i,j}^{\left(t\right)}|}$$

Once the prior parameters are estimated, the residual learning problem Equation (18) can be solved with various sparse reconstruction algorithms [5,6,43]. Firstly, we perform the BM3D denoising operator on $h^{\left(t\right)}$ to get an estimation of the original image. Secondly, we build the low-rank matrix $H_{i}^{\left(t\right)}$, and then transform the data from spatial domain to SVD domain to get the dictionary $D_{i}^{\left(t\right)}$. Note that $H_{i}^{\left(t\right)}$ is the matrix form of $h_{i}^{\left(t\right)}$. Thirdly, using $s_{i}=D_{i}(a_{i}-b_{i})$, we rewrite Equation (18) as
(20)$$s_{i}^{\left(t+1\right)}=\arg\underset{s_{i}}{\min}\frac{{\left\|h_{i}^{\left(t+1\right)}-s_{i}-D_{i}^{\left(t+1\right)}b_{i}^{\left(t+1\right)}\right\|}_{2}^{2}}{2\sigma^{2}}+\sum_{j=1}^{N}\tau_{i,j}^{\left(t+1\right)}|s_{i,j}|$$

Taking a derivative with respect to $s_{i}$, its global optimum is
(21)$$s_{i}^{\left(t+1\right)}=\max({\left(D_{i}^{\left(t+1\right)}\right)}^{T}h_{i}^{\left(t+1\right)}-b_{i}^{\left(t+1\right)}-\sigma^{2}\tau_{i}^{\left(t+1\right)},0)$$
where the noise variance $\sigma^{2}$ is obtained by maximum likelihood estimation [30] (i.e., $\sigma^{2}={\left\|z^{\left(t\right)}\right\|}_{2}^{2}/m$). Using $S_{i}=D_{i}(A_{i}-B_{i})$, $A_{i}$ can be computed by
(22)$$A_{i}^{\left(t+1\right)}={\left(D_{i}^{\left(t+1\right)}\right)}^{T}S_{i}^{\left(t+1\right)}+B_{i}^{\left(t+1\right)}$$

Afterward, we obtain the matrix constructed by similar patches, therefore, $X_{i}^{\left(t+1\right)}=D_{i}^{\left(t+1\right)}A_{i}^{\left(t+1\right)}$, and then recover $x^{\left(t+1\right)}$ by averaging all reconstructed patches. In general, the proposed algorithm is summarized in **Algorithm 1**, named as the D-AMP algorithm with residual learning (RL-DAMP). 


**Algorithm 1. The D-AMP Algorithm with Residual Learning.**
**Input:***y*, *A*, *T*, α, β, *x*^0^ = 0, *z*^0^ = *y*.**For**
*t* = 1 **to**
*T*
**do**(a) Approximate the Onsager correction term via the MC method.(b) Update the residual z(t) with Equation (6).(c) Obtain the intermediate noisy image h(t)=x(t)+A∗z(t), and estimate the noise variance σ2.(d) Perform the denoising operator based on residual learning:   **For**
*i* = 1 **to**
*G*
**do**  (I) Perform the BM3D denoising operator on $h^{\left(t\right)}$ to get the image Db.  (II) Construct the low-rank matrix $H_{i}^{\left(t\right)}$ and perform the SVD on $H_{i}^{\left(t\right)}$ to get the dictionary $D_{i}^{\left(t\right)}$.  (III) Estimate the expectations of scale parameters $\tau_{i,j}^{\left(t+1\right)}$ via Equation (19).  (IV) Compute the global optimums of coefficients $s_{i}^{\left(t+1\right)}$ via Equation (21), and then obtain $A_{i}^{\left(t+1\right)}$ via Equation (22).  (V) Obtain the matrix constructed by similar patches, therefore, $X_{i}^{\left(t+1\right)}=D_{i}^{\left(t+1\right)}A_{i}^{\left(t+1\right)}$.  (VI) If *i* = *G*, recover the whole image $x^{\left(t+1\right)}$ by aggregating all recovered pixels. 

## 4. Experiments

Numerous experiments have been conducted to show the superiority of the proposed CS recovery method RL-DAMP by comparing with image reconstruction algorithms, including three nonlocal sparsity algorithms: NLR-CS [30], BM3D-AMP [31], and the AMP algorithm with low-rank regularization (LR-AMP) [36], and a deep learning algorithm ADMM-Net (the Net with Alternating Direction Method of Multipliers) [25]. The experimental results including objective quality, subjective quality, and runtime are present. Peak Signal-to-Noise Ratio (PSNR) and Structural Similarity Index (SSIM) are used for quantitative evaluation. 

We generated the CS measurements by randomly sampling the Fourier transform coefficients of test images (i.e., *A* is partial Fourier transform with *m* rows and *n* columns). Thus, the sampling ratio was *m*/*n*. We followed the sampling strategy of previous works [12,17], which randomly choose more Fourier coefficients from low frequency and less on high frequency, and set the sampling ratio near to 0.2, as CS imaging is always interested in low-sampling-ratio cases. In our experiments, both the natural images and simulated MR images with size of 256 × 256 or 128 × 128 were used to verify the performance of the proposed CS method, as shown in Figure 3. 

### 4.1. Parameter Settings

For fair comparisons, we offer and analyze the parameters of our algorithms and the comparison algorithms. All codes were downloaded from the authors’ websites. Default settings in their codes were adopted for the reason that parameter optimization has been done by the authors of the original paper. The main parameters of these algorithms were set as follows: (a) Patch size and the number of similar patches for each exemplar patch. In our method, to obtain a dictionary which is a square matrix, the constructed low-rank matrix must be also a square matrix. Therefore, the patch size was set to 6 × 6, and a total of 36 similar patches were selected for each exemplar patch. These two parameters in LR-AMP are the same for comparisons, although the constructed low-rank matrix in LR-AMP must not be a square matrix. These two parameters are 8 × 8, 16 and 6 × 6, 43 in BM3D-AMP and NLR-CS, respectively. We have tried to set these parameters to be the same as the ones of RL-DAMP, but not getting better performance. It is not easy to search similar blocks for too large patch size; thus, these two parameters should be appropriately set up. (b) To reduce the computational complexity, we extracted exemplar image patch in every 5 pixels along both horizontal and vertical directions. In practice, the smaller this parameter the better. It is the same in LR-AMP, and is 3 and 6 pixels in BM3D-AMP and NLR-CS, respectively. (c) For better CS recovery performance, some parameters in our method were tuned empirically, including: the parameters of Gamma distribution α
*=* 0, β
*=* 0.01, as suggested in [38]. There are also empirical parameters in other methods, which can be found in their papers. (d) All nonlocal sparsity algorithms terminated after 50 iterations, except 130 iterations for NLR-CS. All experiments were on a desktop with 3.80GHz AMD A10-5800K CPU. MATLAB version is R2018a. We first present the experimental results for noiseless CS measurements, and then report the results using noisy CS measurements. The noisy CS measurements are mixed with Gaussian white noise with standard deviation 8 or 15.

### 4.2. Experiments on Noiseless Data

To reduce the randomness, all results were recorded by averaging the values after repeating each experiment five times for each image. The PSNR and SSIM comparison results of recovered images by competing CS recovery methods are shown in Figure 4. From Figure 4, one can see that the highest PSNR and SSIM results were achieved by the proposed algorithm RL-DAMP. In fact, the average PSNR gain over BM3D-AMP, LR-AMP, ADMM-Net and NLR-CS can be as much as 0.34 dB, 0.55 dB, 0.99 dB, and 1.25 dB. The PSNR and SSIM curves declined with the decrease in sampling ratio. RL-DAMP always performed better at all sampling ratio. Besides, the proposed algorithm RL-DAMP outperformed the LR-AMP algorithm, which validates the effectiveness of our residual learning strategy. Note that the LR-AMP algorithm is similar to the RL-DAMP method without using the residual learning strategy, which solves the following objective function:(23)$${\hat{A}}_{i}=\underset{A_{i}}{\arg\min}{\left\|H_{i}-D_{i}A_{i}\right\|}_{F}^{2}+\lambda {\left\|D_{i}A_{i}\right\|}_{*}$$

By comparing RL-DAMP with LR-AMP in the experiments, we can analyze the effect of our residual learning strategy. 

We can compare three D-AMP related algorithms: RL-DAMP, BM3D-AMP, and LR-AMP in terms of steps, penalties, complexity, and convergence, and then validate them in our following experiments. Three D-AMP related algorithms all have two steps: a denoising and a residual update step. The sparse penalties of RL-DAMP and LR-AMP can be shown in Equations (7) and (23), respectively, while BM3D-AMP, which uses the BM3D denoiser as the implicit sparse terms, does not have explicit penalties. Note that, these sparse penalties play a role of the denoising step in the general steps of the D-AMP-related algorithms. Recall that, *G* is the total number of similar patch groups, the size of the low-rank matrices is M×N, where *M* is the number of similar patches, and the size of an image patch is $\sqrt{N}\times \sqrt{N}$. The most time-consuming operation in our method is that performing the SVD on the low-rank matrix to get the dictionary. The complexity of this step is $O(GM^{3})$. This step is also the most time-consuming operation of LR-AMP; thus, its complexity is also $O(GM^{3})$. The most time-consuming operation of BM3D-AMP is that performing the three-dimensional (3D) wavelet transform on the 3D image cube. The complexity of this step is $O(G{\sqrt{N}}^{3})$. Note that *M* ≥ *N*, thus, the complexity of RL-DAMP and LR-AMP is higher than the one of BM3D-AMP. As the BM3D method is written in C language, the running time of the BM3D-AMP method is much shorter. These three methods are based on the D-AMP algorithm and inherit its convergence. The convergence can be validated in the following experiments with the iteration number vs. PSNR curves and CPU (central processing unit) time vs. PSNR curves.

We classify the test images into two sets: the set of magnetic resonance (MR) images and the set of natural images. The PSNR comparison results of these two image sets by competing CS recovery methods are shown in Figure 5. From Figure 5, one can see that: (1) the PSNR curves of all algorithms on MR images are higher than the ones on natural images, for the reason that the structure of MR images is much more regular; (2) the proposed algorithm RL-DAMP always performed better than other reconstruction methods for both MR images and natural images, which implies that the assumption of the nonlocal sparsity structure is appropriate for both MR images and natural images; (3) the deep learning method ADMM-Net achieved better PSNR results on MR images than natural images because it uses the MR images as the training samples, which is less appropriate for natural images with much more irregular structures.

To facilitate the evaluation of subjective qualities, Figure 6 and Figure 7 show the visual comparisons of the reconstructed results on Boat and Chest image with 20% sampling by different methods respectively, while the corresponding iterative curves are given in Figure 8 and Figure 9, respectively. From Figure 6 and Figure 7, we can clearly see that the proposed algorithm RL-DAMP performed better than others, which enjoyed great advantages in producing clearer images, for example, on the area of ropes and beach in Figure 6, and blood vessels in Figure 7. It could not only perfectly reconstruct large-scale sharp edges, but also well recover small-scale fine structures. The images reconstructed by the LR-AMP and BM3D-AMP methods were over-smoothed. These two methods have a strong assumption of the nonlocal self-similarity structure, however, many images with irregular structures do not strictly follow this assumption. With the use of residual learning, the RL-DAMP algorithm can overcome this issue. Visual artifacts can still be as clearly observed on the image produced by the ADMM-Net method, especially for natural images, which implies that the generalization ability of the algorithm need to enhance. The visual results recovered by the NLR-CS algorithm were always inferior to RL-DAMP, with the lower PSNR or SSIM values and less details. We could find that the RL-DAMP algorithm was the closest to the original with only 20% sampling ratio, which had great superiority on the images with irregular structures because of the residual learning. The superiority of the proposed algorithm in visual quality could be demonstrated by these results.

The CPU time and PSNR are traced in each iteration for each of reconstruction methods. Figure 8a (or Figure 9a) and Figure 8b (or Figure 9b) present the iteration number vs. PSNR curves and CPU time vs. PSNR curves, respectively. In the case of NLR-CS with more iterations, its results are every two iterations, presented in our figures. Although residual learning definitely cost a little more time to solve, which made RL-DAMP slower than LR-AMP, it achieved the best performance in terms of PSNR and CPU time after about 20 iterations, in Figure 8a and Figure 9a, and after about 150 s, in Figure 9b and Figure 10b. These curves demonstrate that RL-DAMP can converge to a good reconstructed result in a reasonable amount of time and can reduce requirement for the number of measurements or increase the accuracy of the solution with the same measurements. The RL-DAMP algorithm was relatively slow. The main computational burdens were introduced by iteratively applying SVD on each patch group. The deep learning method ADMM-Net was the fastest one, which took only several seconds to reconstruct an image, but it spent a lot of time to train a deep learning network.

### 4.3. Experiments on Noisy Data

Considering in realistic settings compressive samples are subject to measurement noise, we conducted similar experiments with noisy CS measurements to demonstrate the robustness of the proposed RL-DAMP to noise. Noisy sampling can be modeled by y=Ax+w, where *w* represents additive white Gaussian noise (AWGN). The standard deviations of AWGN 8 and 15 were used in our experiments, representing the environment with low noise and high noise, respectively. In Figure 10, we compare the performance of the proposed RL-DAMP to other CS reconstructed methods when varying amounts of measurement noise were present. In Figure 11, Figure 12, Figure 13 and Figure 14, we provide visual comparisons between the reconstructions in the presence of measurement noise. The corresponding iteration number vs. PSNR curves and CPU time vs. PSNR curves are present in Figure 11 and Figure 12, which show that all algorithms converged to reconstructed results quickly. The trends of PSNR comparisons in these figures are consistent with previous experiments. But the quality of the images reconstructed by all CS methods degraded seriously as measurement noise increased. The RL-DAMP method was found to be exceptionally robust to noise in the experiments. Because it is based on the denoising-based approximate message passing algorithm, which regards the regularization operation as a denoising process. When these noises exited in the process of CS measurement, the amplitude of noises, which was relevant to the threshold of regularization (see (21)) could be estimated by the proposed algorithm, thus, our method can reduce the noise effectively.

### 4.4. Experiments on Small-Size Images

To fully demonstrate the performance of the proposed algorithm, we also tested our method on small-size images. We resized all images to 128 × 128. Since the net of the deep learning method ADMM-Net was trained for the image with size of 256 × 256, it cannot be run on images with other sizes. Thus, this experiment did not include the ADMM-Net method. Figure 15 and Figure 16 show the visual comparisons of the reconstructed results on Barbara image with 20% sampling by different methods on noiseless data and noisy data, respectively. From Figure 15 and Figure 16, we can clearly see that the proposed algorithm RL-DAMP performed better than others on both noiseless data and noisy data, which is consistent with the experiments on images with size of 256 × 256. Our method produced clearer images with more small-scale fine structures (e.g., on the area of background and headscarf) and achieved the best PSNR and SSIM results. The superiority of the proposed algorithm on small-size image could be demonstrated by these results. We also provide the sampling mask image at 20% sampling ratio with these two sizes in Figure 17, from which we can see that our sampling strategy randomly chose more Fourier coefficients from low frequency and less on high frequency. It is consistent with the sampling strategy in previous works [12,17].

## 5. Conclusions

In this paper, we have presented a new entropy-based approach toward image CS reconstruction based on nonlocal residual learning and denoising-based approximate message passing algorithm. Nonlocal residual learning enables us to enhance the group sparsity of similar patches; denoising-based approximate message passing algorithm in the frame of the EM method offers a principled and computationally efficient solution to image reconstruction from recovered patches. By taking the scale parameter of the LSM model as a random variable, it makes the practical representation much more feasible for achieving a better spatial adaptation. When compared against existing CS techniques, our RL-DAMP method is mostly favored in terms of both subjective and objective qualities. Our simulations and experiments on a variety of images demonstrate the superiority of the proposed algorithm to several nonlocal sparsity-based algorithms or solvers and a deep learning method in CS image recovery. Moreover, it shows significant performance improvements over a wide range of images, including natural and MR ones. How the nonlocally regularized image reconstruction algorithm jointly works with the deep learning method deserves further study.

## Figures and Tables

**Figure 1 entropy-21-00900-f001:**
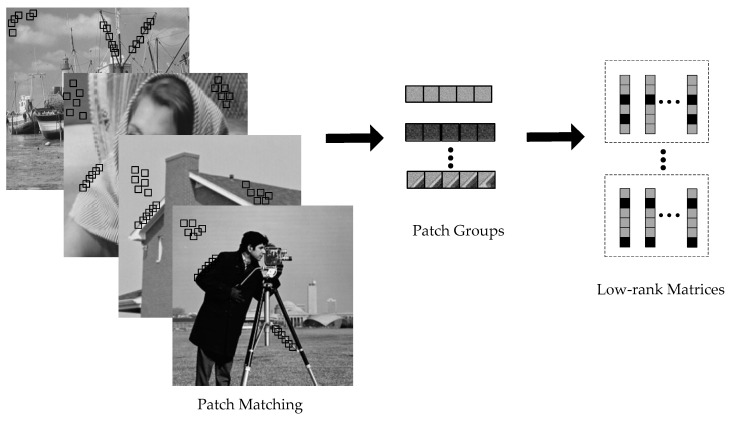
Illustrations for the low-rank matrices construction.

**Figure 2 entropy-21-00900-f002:**
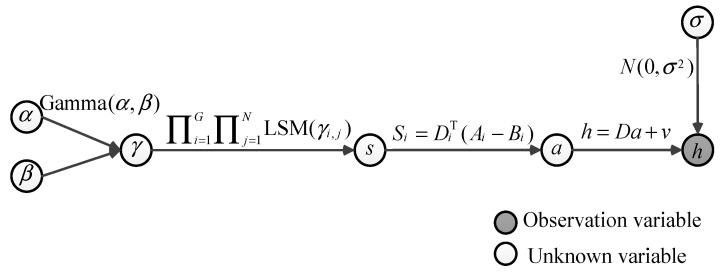
The hierarchical signal model.

**Figure 3 entropy-21-00900-f003:**
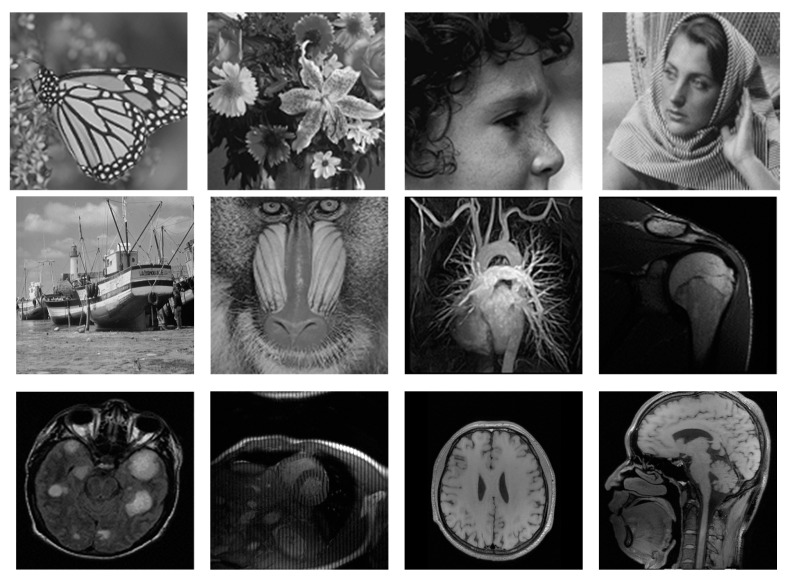
Test images used for compressive sensing experiments.

**Figure 4 entropy-21-00900-f004:**
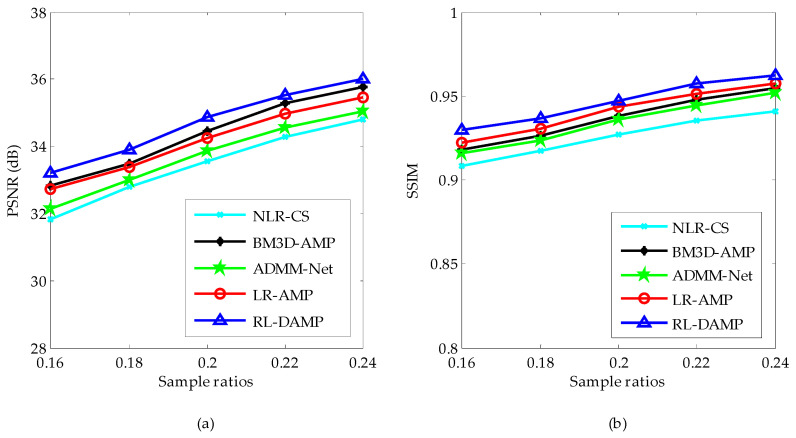
Average Peak Signal-to-Noise Ratio (PSNR) (**a**) and Structural Similarity Index (SSIM) (**b**) at different sample ratios.

**Figure 5 entropy-21-00900-f005:**
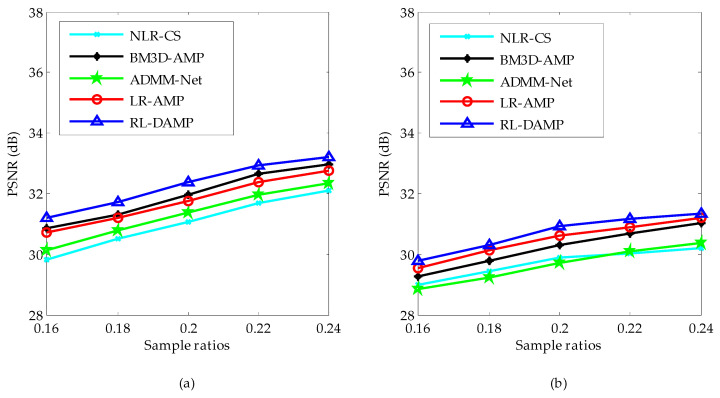
Average PSNR on magnetic resonance (MR) images (**a**) and natural images (**b**).

**Figure 6 entropy-21-00900-f006:**
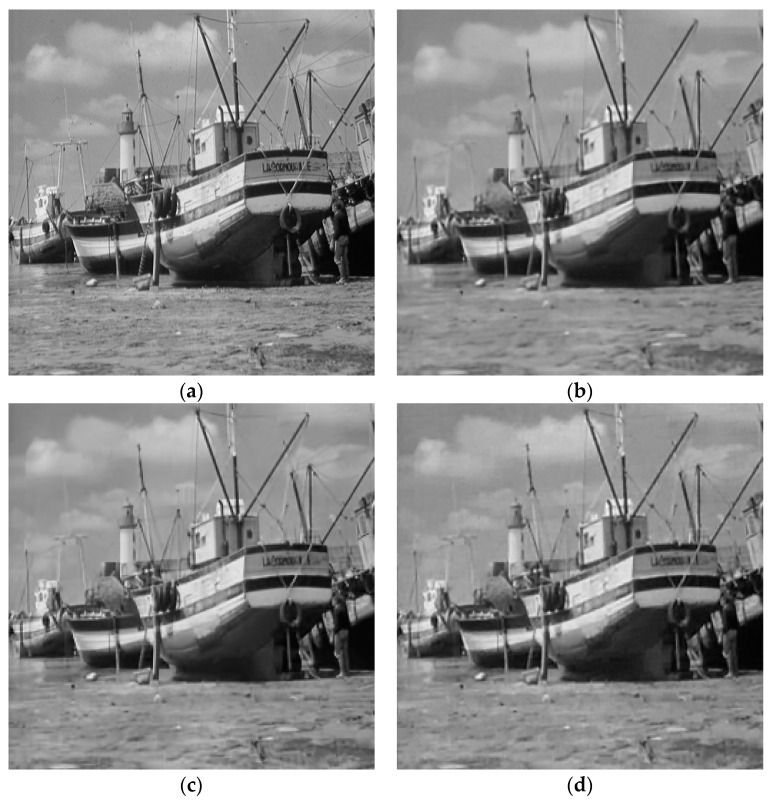
Visual comparisons on Boat image at 20% sampling ratio. (**a**) The original; (**b**) NLR-CS (PSNR: 32.08 dB, SSIM: 0.9068); (**c**) BM3D-AMP (PSNR: 32.29 dB, SSIM: 0.9387); (**d**) ADMM-Net (PSNR: 32.71 dB, SSIM: 0.9226); (**e**) LR-AMP (PSNR: 31.92 dB, SSIM: 0.9435); (**f**) RL-DAMP (PSNR: 33.92 dB, SSIM: 0.9473).

**Figure 7 entropy-21-00900-f007:**
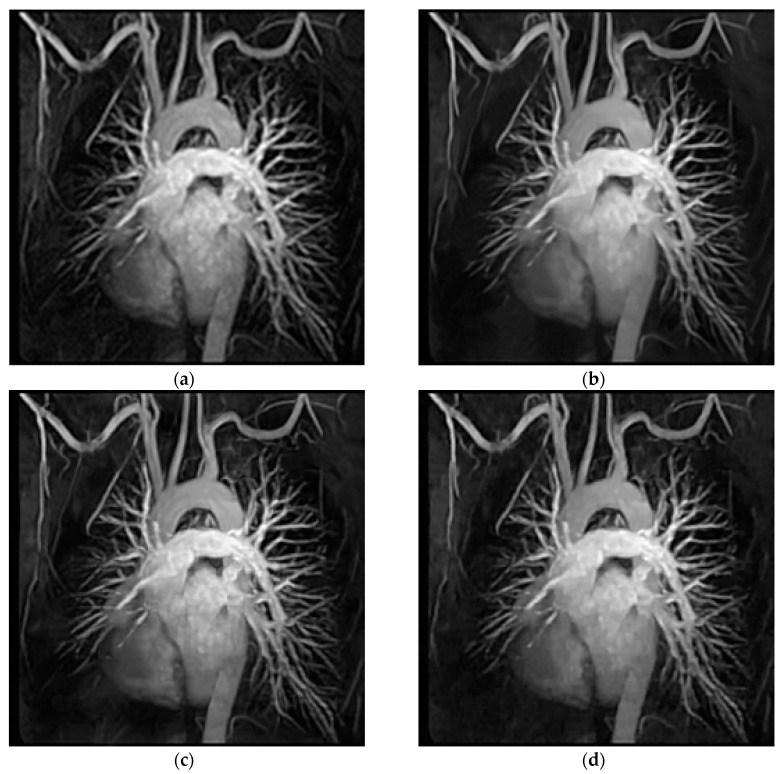
Visual comparisons on Chest image at 18% sampling ratio. (**a**) The original; (**b**) NLR-CS (PSNR: 30.80 dB, SSIM: 0.9065); (**c**) BM3D-AMP (PSNR: 31.70 dB, SSIM: 0.9353); (**d**) ADMM-Net (PSNR: 32.33 dB, SSIM: 0.9187); (**e**) LR-AMP (PSNR: 31.05 dB, SSIM: 0.9353); (**f**) RL-DAMP (PSNR: 32.57 dB, SSIM: 0.9435).

**Figure 8 entropy-21-00900-f008:**
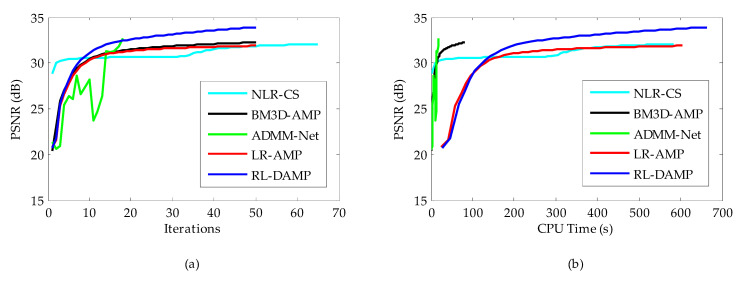
Iterative curves on Boat image at 20% sampling ratio. (**a**) Average PSNR to iterations; (**b**) Average PSNR to CPU (central processing unit) time.

**Figure 9 entropy-21-00900-f009:**
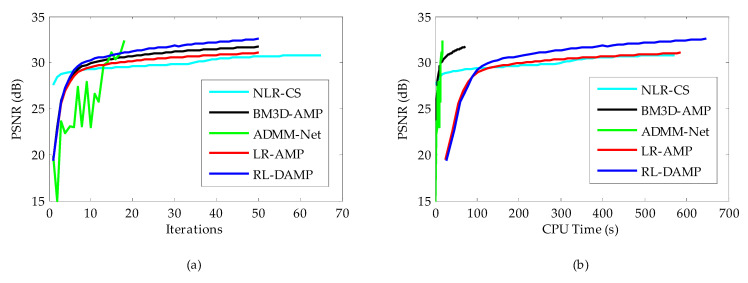
Iterative curves on Chest image at 20% sampling ratio. (**a**) Average PSNR to iterations; (**b**) Average PSNR to CPU time.

**Figure 10 entropy-21-00900-f010:**
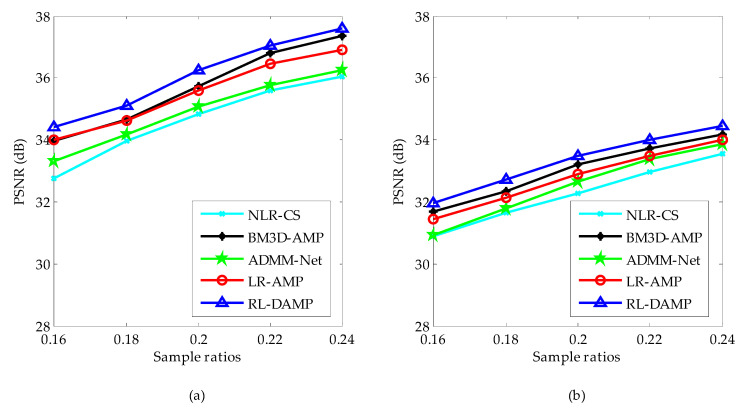
Average PSNR at different sample ratios with measurement noise with standard deviation 8 and 15. (**a**) The standard deviation is 8; (**b**) the standard deviation is 15.

**Figure 11 entropy-21-00900-f011:**
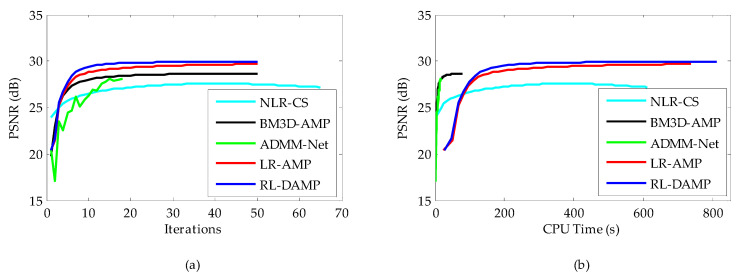
Iterative curves on Barbara image at 20% sampling ratio with measurement noise with standard deviation 15. (**a**) Average PSNR to iterations; (**b**) average PSNR to CPU time.

**Figure 12 entropy-21-00900-f012:**
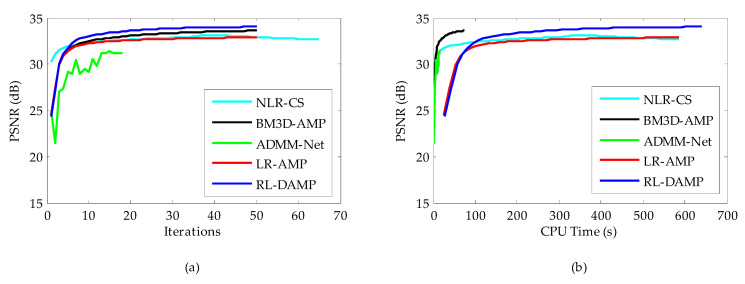
Iterative curves on Brain image at 20% sampling ratio with measurement noise with standard deviation 8. (**a**) Average PSNR to iterations; (**b**) average PSNR to CPU time.

**Figure 13 entropy-21-00900-f013:**
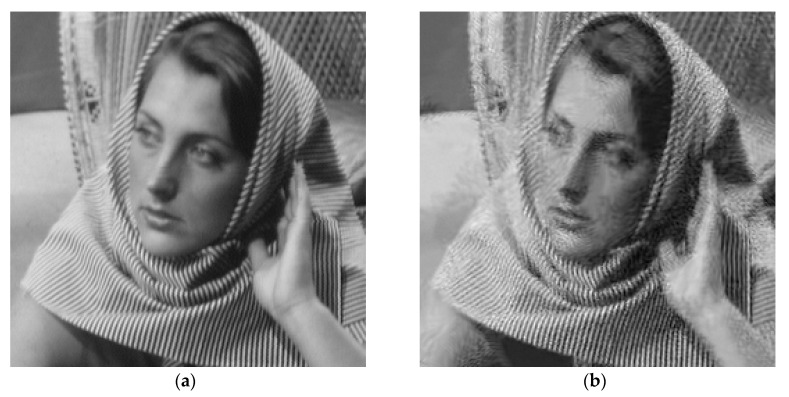
Visual comparisons on Barbara image at 20% sampling ratio with measurement noise with standard deviation 15. (**a**) The original; (**b**) NLR-CS (PSNR: 27.16 dB, SSIM: 0.8061); (**c**) BM3D-AMP (PSNR: 28.65 dB, SSIM: 0.8465); (**d**) ADMM-Net (PSNR: 28.13 dB, SSIM: 0.8270); (**e**) LR-AMP (PSNR: 29.69 dB, SSIM: 0.8703); (**f**) RL-DAMP (PSNR: 29.89 dB, SSIM: 0.8794).

**Figure 14 entropy-21-00900-f014:**
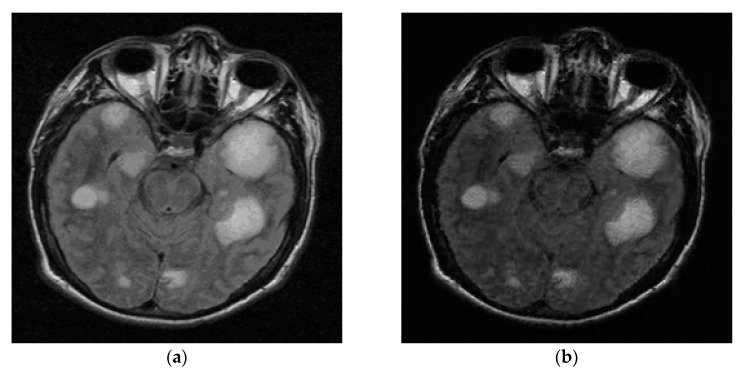
Visual comparisons on Brain image at 20% sampling ratio with measurement noise with standard deviation 8. (**a**) The original; (**b**) NLR-CS (PSNR: 32.65 dB, SSIM: 0.8714); (**c**) BM3D-AMP (PSNR: 33.63 dB, SSIM: 0.8962); (**d**) ADMM-Net (PSNR: 31.21 dB, SSIM: 0.8107); (**e**) LR-AMP (PSNR: 32.93 dB, SSIM: 0.8743); (**f**) RL-DAMP (PSNR: 34.05 dB, SSIM: 0.8976).

**Figure 15 entropy-21-00900-f015:**
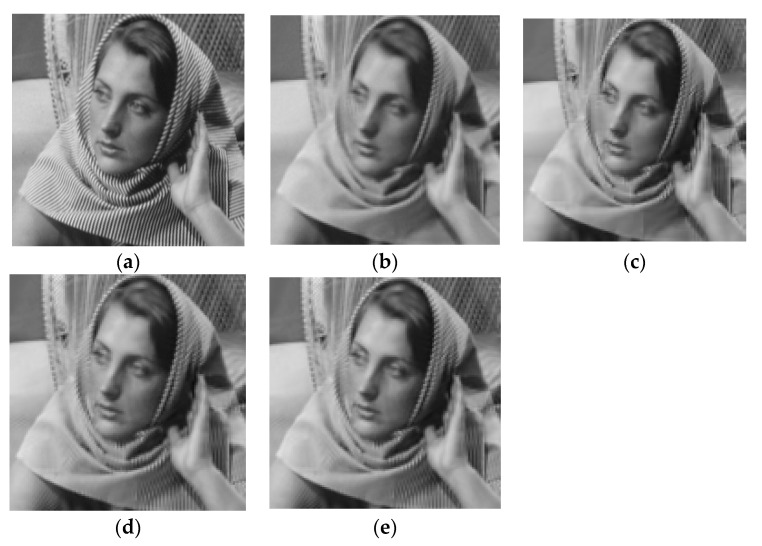
Visual comparisons on Barbara image at 20% sampling ratio with size of 128 × 128. (**a**) The original; (**b**) NLR-CS (PSNR: 29.18 dB, SSIM: 0.8890); (**c**) BM3D-AMP (PSNR: 28.49 dB, SSIM: 0.8997); (**d**) LR-AMP (PSNR: 29.73 dB, SSIM: 0.9233); (**e**) RL-DAMP (PSNR: 29.94 dB, SSIM: 0.9240).

**Figure 16 entropy-21-00900-f016:**
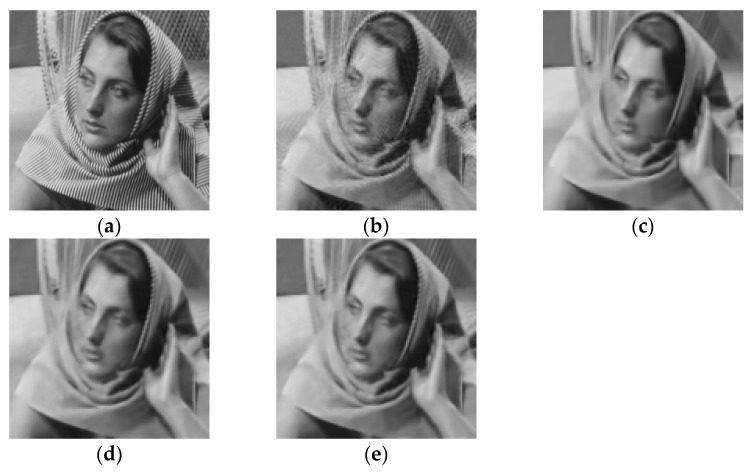
Visual comparisons on Barbara image at 20% sampling ratio with size of 128 × 128 with measurement noise with standard deviation 8. (**a**) The original; (**b**) NLR-CS (PSNR: 27.17 dB, SSIM: 0.8369); (**c**) BM3D-AMP (PSNR: 27.01 dB, SSIM: 0.8366); (**d**) LR-AMP (PSNR: 27.74 dB, SSIM: 0.8486); (**e**) RL-DAMP (PSNR: 27.93 dB, SSIM: 0.8495).

**Figure 17 entropy-21-00900-f017:**
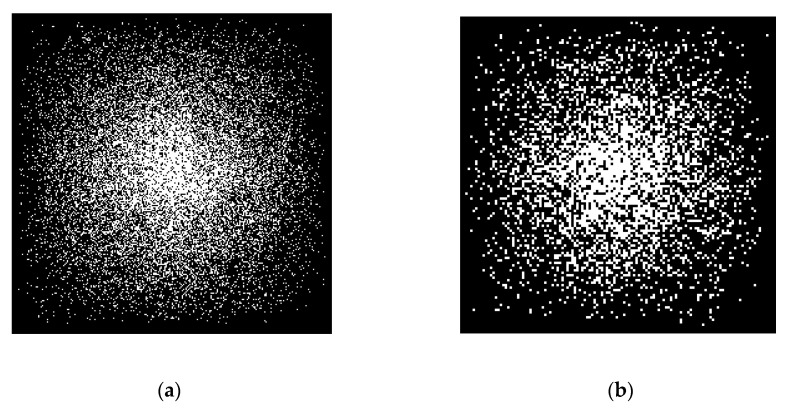
The sampling mask at 20% sampling ratio with different sizes. (**a**) 256 × 256; (**b**) 128 × 128.

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
