# Peer review of "An Entropy-Based Algorithm with Nonlocal Residual Learning for Image Compressive Sensing Recovery"

_entropy, 2019, doi:10.3390/e21090900_

Round 1

Reviewer 1 Report

An Entropy-Based Algorithm with Nonlocal Residual Learning for Image Compressive Sensing Recovery

In general it is a well written paper with clear structure. It combines several exiting techniques including entropy based algorithm, nonlocal residual learning using sparsity, laplacian mixture mixture, D-AMP, ect. Here are some comments.

The language needs to be improved further.

Paper with sparsity in total variation, such as [1-2] were missing in the literature review.

It says (7) can be easily solved by (9), any reference? And how can (7) be solved by D-AMP? More explanations are required for readers.

The authors should show that the proposed method that combines several approaches are necessary for the specific problem they are dealing with, rather than a simple concatenation of several existing approaches arbitrarily.

A table to compare D-AMP algorithm with residual learning and other D-AMP related algorithms is necessary, in terms of steps, penalties, complexity, convergence, so on.

Are other algorithms’ parameters set as optimal in the comparison in simulation?  

Please provide more figure comparisons with smaller sizes

Are the codes used in experiments available on line?

[1] Shi, Feng, Jian Cheng, Li Wang, Pew-Thian Yap, and Dinggang Shen. "LRTV: MR image super-resolution with low-rank and total variation regularizations." IEEE transactions on medical imaging 34, no. 12 (2015): 2459-2466.

[2] Li, Kezhi, Thusara C. Chandrasekera, Yi Li, and Daniel J. Holland. "A non-linear reweighted total variation image reconstruction algorithm for electrical capacitance tomography." IEEE Sensors Journal 18, no. 12 (2018): 5049-5057.

The paper needs to be revised carefully before going to the next step.

Reviewer 2 Report

The following is my review of the manuscript, "An Entropy-Based Algorithm with Nonlocal Residual Learning for Image Compressive Sensing Recovery”-Entropy-589364. The technical components of the paper are sound and the manuscript proposes “a novel entropy-based algorithm for CS recovery to enhance image sparsity through leaning the group sparsity of (the) residual. To reduce the residual of similar packed patches, the group sparsity of (the) residual

is described by a Laplacian scale mixture (LSM) model, i.e., each singular value of the residual of similar packed patches is modeled as a Laplacian distribution with a variable scale parameter, (to) exploit the benefits of high-order dependency among sparse coefficients.”

The paper contains some interesting ideas and results. It may hold some promise. While there were several references to related material, I would suggest that the author(s) incorporate the following reference that would add to the literature and take into consideration additional entropy-based algorithms that may enhance their study:

James A. Rodger. QuantumIS: A Qualia Consciousness Awareness and Information Theory Quale Approach to Reducing Strategic Decision-Making Entropy. Entropy 2019, 21, 125; doi:10.3390/e21020125 doi: ww.mdpi.com/journal/entropy

Further, the article needs to be edited for grammar and flow. There are many missing articles such as “a” “an” and “the” that detract from the readability of the document. Therefore, the author(s) must try to make their discussion a bit more apparent, with an expanded and more clear, logical flow.

Hopefully, these comments don’t come across as overly negative as the paper possess a good deal of potential. I hope that the comments and feedback will prove useful as the author(s) continue to develop and revise this manuscript. The previously mentioned strengths of the manuscript suggest that the paper can be considered to be of high quality and also be of very sufficient value to the readership of "Entropy”. Therefore, based on the foregoing discussion, I would recommend that this paper be accepted, after a thorough editing and after incorporating the previously mentioned citation and related concepts and suggestions into the manuscript.

Reviewer 3 Report

Overall:

This paper described an entropy based compressive sensing approach for image reconstruction using nonlocal residual learning and denoising-based approximate message passing algorithm. Overall, the proposed method may have some merits, but there are still some major concerns about this work. Please also check the writing styles and English thoroughly.

Detailed comments:

Are there any statistical differences among the different comparison methods. For the evaluation of the reproducibility, the Matlab implementation should be published on Github. How reliable is the proposed method? The authors should show the undersampling pattern. Recently, deep learning based compressive sensing have been used in many applications. The novelty of the proposed method seems limited. The following papers must be cited for the state-of-the-art comparison:

Seitzer, Maximilian, et al. "Adversarial and perceptual refinement for compressed sensing MRI reconstruction." International Conference on Medical Image Computing and Computer-Assisted Intervention. (pp. 232-240), 2018.

Schlemper, Jo, et al. "Stochastic Deep Compressive Sensing for the Reconstruction of Diffusion Tensor Cardiac MRI." International Conference on Medical Image Computing and Computer-Assisted Intervention. (pp. 295-303), 2018.

The English writing of the manuscript must be checked thoroughly. There are lots of writing style problems, for example:

‘When these exits noises in the process of CS measurement, the propose algorithm can estimate the amplitude of noises, then automatically adjusts the threshold of regularization to reduce the noise effectively.’ The sentence needs to be rephrased.

Round 2

Reviewer 1 Report

The revised paper does not address the reviewer’s concerns well. It does not meet the standard of a qualified paper in the journal. Specifically, the problems are:

The novelty of the paper is limited, considering nuclear norm penalty, soft thresholding, and D-Amp are widely used. Many important references in the field are missing. The revised paper failed to provide a detailed comparison with regards to the steps, penalties, complexity, convergence analysis to other methods The comparison algorithms are using the default parameters, which are far from what they can really achieve.

The paper has to be revised thoroughly and address the above concerns carefully.

Reviewer 3 Report

Overall:

I am not satisfied with the author’s feedback.

Detailed comments:

Regarding my first question: ‘For the evaluation of the reproducibility, the Matlab implementation should be published on Github.’

The authors mentioned ‘we want to come up with something new’. This is not a reasonable excuse because some basic implementation can be published. The latest development can be done later by the authors for their further publications.

The authors also mentioned ‘To be honest, it is very easy to implement our method.’ Again, the ‘easy version’ of the implementation should be opensource for reproducibility test. Not because it is easy then can be hided.

Regarding my second question ‘How reliable is the proposed method? The authors should show the undersampling pattern.’ The authors have done something wrong. For the 2D MRI data, the frequency encoding direction is always very fast and doesn’t need undersampling to accelerate. The applied 2D Gaussian undersampling mask therefore is not reasonable. The experiments are flawed and not realistic. The authors should perform experiments using 1D undersampling masks. The following references must be included for discussions.

Yang, Guang, et al. "DAGAN: deep de-aliasing generative adversarial networks for fast compressed sensing MRI reconstruction." IEEE transactions on medical imaging 37.6 (2018): 1310-1321.

Yu, Simiao, et al. "Deep de-aliasing for fast compressive sensing MRI." arXiv preprint arXiv:1705.07137 (2017).

For comparison methods, the authors mentioned that other methods might implemented in Python and can’t be compared, which is also very unreasonable. For example, the following work has opensource implementation online, the authors should be able to test with different undersampling masks and run the Python codes easily. It requires nothing but changing the input of the program.

Schlemper, Jo, et al. "Stochastic Deep Compressive Sensing for the Reconstruction of Diffusion Tensor Cardiac MRI." International Conference on Medical Image Computing and Computer-Assisted Intervention. (pp. 295-303), 2018.

Round 3

Reviewer 1 Report

The paper has been improved properly. The only concern is that if the comparisons are fair of other algorithms using the default settings. Others may use an existing algorithm and achieve a better result by using appropriate parameters. Thus please summarize and list crucial parameters that used in Fig. 10 to Fig. 12 in a table in the appendix for the purpose of completeness.

Reviewer 3 Report

The authors have done considerable changes. Although not absolutely satisfied, I believe the paper might be accepted before some minor corrections. For the literature review, the authors should also added this paper because it is an open source project with uncertainty measurement of the MRI reconstruction that the authors can add into future comparison studies.

Schlemper, Jo, et al. "Stochastic Deep Compressive Sensing for the Reconstruction of Diffusion Tensor Cardiac MRI." International Conference on Medical Image Computing and Computer-Assisted Intervention. (pp. 295-303), 2018.
